# Post-Acquisition Hyperpolarized ^29^Silicon Magnetic Resonance Image Processing for Visualization of Colorectal Lesions Using a User-Friendly Graphical Interface

**DOI:** 10.3390/diagnostics12030610

**Published:** 2022-03-01

**Authors:** Caitlin V. McCowan, Duncan Salmon, Jingzhe Hu, Shivanand Pudakalakatti, Nicholas Whiting, Jennifer S. Davis, Daniel D. Carson, Niki M. Zacharias, Pratip K. Bhattacharya, Mary C. Farach-Carson

**Affiliations:** 1Department of Electrical and Computer Engineering, Rice University, Houston, TX 77005, USA; cvm1@rice.edu (C.V.M.); drs3@rice.edu (D.S.); 2Department of Diagnostic and Biomedical Sciences, School of Dentistry, The University of Texas Health Science Center, Houston, TX 77054, USA; 3Department of Bioengineering, Rice University, Houston, TX 77005, USA; hujingzhe@pm.me; 4Department of Cancer Systems Imaging, The University of Texas MD Anderson Cancer Center, Houston, TX 77030, USA; spudakalakatti@mdanderson.org (S.P.); whitingn@rowan.edu (N.W.); pkbhattacharya@mdanderson.org (P.K.B.); 5Department of Epidemiology, The University of Texas MD Anderson Cancer Center, Houston, TX 77030, USA; jdavis57@kumc.edu; 6Department of BioSciences, Rice University, Houston, TX 77005, USA; dcarson@rice.edu; 7Department of Urology, The University of Texas MD Anderson Cancer Center, Houston, TX 77030, USA; nmzacharias@mdanderson.org

**Keywords:** colorectal cancer, diagnostic imaging, hyperpolarization, image processing, GUI, MRI, silicon particles

## Abstract

Medical imaging devices often use automated processing that creates and displays a self-normalized image. When improperly executed, normalization can misrepresent information or result in an inaccurate analysis. In the case of diagnostic imaging, a false positive in the absence of disease, or a negative finding when disease is present, can produce a detrimental experience for the patient and diminish their health prospects and prognosis. In many clinical settings, a medical technical specialist is trained to operate an imaging device without sufficient background information or understanding of the fundamental theory and processes involved in image creation and signal processing. Here, we describe a user-friendly image processing algorithm that mitigates user bias and allows for true signal to be distinguished from background. For proof-of-principle, we used antibody-targeted molecular imaging of colorectal cancer (CRC) in a mouse model, expressing human MUC1 at tumor sites. Lesion detection was performed using targeted magnetic resonance imaging (MRI) of hyperpolarized silicon particles. Resulting images containing high background and artifacts were then subjected to individualized image post-processing and comparative analysis. Post-acquisition image processing allowed for co-registration of the targeted silicon signal with the anatomical proton magnetic resonance (MR) image. This new methodology allows users to calibrate a set of images, acquired with MRI, and reliably locate CRC tumors in the lower gastrointestinal tract of living mice. The method is expected to be generally useful for distinguishing true signal from background for other cancer types, improving the reliability of diagnostic MRI.

## 1. Introduction

A growing number of sophisticated medical imaging technologies are available for cancer detection and diagnosis, but there remains a need for accurate, user-friendly analysis of the images obtained. Colorectal cancer (CRC), the third leading cause of cancer-related death, has most deaths attributable to late diagnosis after the cancer has invaded the intestinal wall and accessed the venous system [1,2,3]. Screening is of critical importance because early-stage CRC is often asymptomatic, but if caught early, patients have much higher survival rates [4,5]. The standard of care for screening is colonoscopy, but many patients are averse to this procedure and certain patients, such as the elderly, are prone to intestinal perforation and septic infection [6]. Visually assessed colonoscopy is prone to human error because the diagnostician must detect any suspicious lesions by eye, with certain types of polyps, such as flat and depressed polyps, being more difficult to see than others. The location of some polyps, including when diverticulitis is present, can obscure them from visual detection [7], leading to false negatives, allowing cancerous lesions to progress without treatment. Other CRC detection techniques, including computed tomography (CT)-based virtual colonoscopy, provide alternatives to colonoscopy, but suffer from the possibility of false readings, ascribed to the user performing the reading more so than the image capture [8]. More objective, effective and accessible image analysis tools, as well as patient-friendly screening methods, can improve the accuracy of diagnosis, eliminate false readings, and improve patient outcomes.

Magnetic resonance imaging (MRI) is noninvasive and, unlike CT, does not utilize ionizing radiation. Hyperpolarization through dynamic nuclear polarization (DNP) is an emerging method to increase the signal intensity and duration of magnetic resonance (MR) active nuclei for medical imaging applications [9]. Hyperpolarization boosts the measured signal in MRI by bypassing the room temperature thermodynamic distribution of nuclear spin states, improving signal to noise and increasing sensitivity. Commonly utilized isotopes for hyperpolarized MRI are ^13^C, ^15^N, and ^129^Xe [9]. MRI, used to image human anatomy since the 1970s [10], utilizes the nuclear spin of hydrogen to acquire anatomical information that is directly related to the variation of water content among tissue types [10]. Acquisition time for MRI can be considerably long, but it provides a means to perform whole body and deep tissue imaging without ionizing radiation in vulnerable patients [11,12].

A promising new experimental screening method for polyp detection is targeted molecular imaging with MRI, using hyperpolarized silicon nanoparticles that are introduced into the colon and specifically bind to cancerous lesions where they can be detected. We previously described the use of antibody-functionalized hyperpolarized silicon to image CRC lesions in a humanized mouse model, expressing mucin 1 (MUC1), a cancer biomarker [13]. Limitations of MR imaging analysis are that it (1) typically relies on commercial software that allows for individual imaging adjustments but has limited capabilities for comparative analysis; (2) is susceptible to thresholding difficulties that make it challenging to distinguish true signal from background, especially in an anatomically complex tissue, such as the colon. This can lead to false results that negatively impact diagnosis and treatment planning, including rescans to assess progression and/or recurrence. Hyperpolarized MR imaging can be susceptible to artifacts, which can present differently depending on the acquisition method, pulse sequences, gradients, MR coil and the extent of hyperpolarization buildup. Common artifacts associated with hyperpolarized MRI are motion, aliasing, and radio frequency (RF) overflow artifacts [14,15,16]. Furthermore, due to experimental set up, signal that is not of interest can contribute to the displayed image, skewing the perceived intensity.

Here, we describe a user-friendly, post-acquisition image processing method that considers relative signal intensity to provide an unbiased comparative analysis of MRI imaging data. Using a set of independently acquired images from the humanized MUC1 mouse model, we could co-register the targeted silicon signal with the anatomical proton-acquired image to locate the MUC1-positive lesions. Noise reduction and thresholding methods were integrated into a graphical user interface (GUI) that processes imaging data such that co-registration and comparative analysis can be readily accomplished on site.

## 2. Materials and Methods

### 2.1. Mouse Model for Colorectal Cancer

All procedures involving animals were conducted using approved protocols and under the supervision of the Institutional Animal Care and Use Committee at the University of Texas M.D. Anderson Cancer Center (00000925-RN02, initially approved 31 January 2014, MD Anderson IACUC). Previously, a mouse model was genetically engineered to produce CRC tumors that express human MUC1 on their cell surfaces [17,18]. These mice were obtained from The Jackson Laboratory (stock no. 024631) and crossed with Apc^min/+^ mice (The Jackson Laboratory, stock no. 002020) [19,20]. In order to generate the mice for this study, female mice carrying hemizygous huMUC1 were mated with male Apc^min/+^ mice (heterozygous). MUC1 was identified using a specific, validated IgG1 antibody 214D4 (Sigma-Aldrich, St. Louis, MO, USA). Every two weeks, colonoscopy was performed on the mice to monitor tumor growth as well as the relative location within the colon. Tumors typically ranged in size between 5–10 mm. This protocol was performed on four cohorts of mice with *n* = 3 for each group. The MUC1+ targeted group served as the experimental group.

### 2.2. MR Imaging of CRC Tumors

After tumors were large enough to confirm their presence by colonoscopy, mice were selected for imaging. Thirty minutes before imaging, mice were anesthetized, and an enema was performed to evacuate the bowel for the unimpeded introduction of particles. In addition to the MUC1+ experimental group, there were three control groups: biological control, chemical control, and pre-blocked MUC1+ mice. The biological control group produced tumors that did not express human MUC1 and were imaged using 214D4 functionalized nanoparticles. The chemical control group produced human MUC1 tumors and were imaged using particles coated only with polyethylene glycol (PEG). The pre-blocked mice, which produced tumors that expressed human MUC1, were given an enema and flushed with fluorescent dye-labeled antibody at least 30 min prior to imaging. After allowing the antibody to bind to the MUC1 present in the colon, particles were introduced, and imaging took place as with the other cohorts. Fluorescent imaging before MRI confirmed the successful binding of the blocking agent.

Mice were then placed in a 7 T preclinical scanner (Bruker, Billerica, MA, USA) using a dual-tuned ^1^H/^29^Si coil for co-registered imaging. Hyperpolarized silicon particles (Alfa Aesar, Haverhill, MA, USA) were then taken from the house made dynamic nuclear polarizer (DNP) device described previously [21] and quickly warmed in hand before dissolution in 300 μL of phosphate buffered saline (PBS). The solution of nanoparticles and PBS were administered rectally through a modified syringe using a non-magnetic flexible gavage needle. Immediately following administration of the nanoparticles, a ^29^Si MR spectra (α = 10°) was acquired to observe the initial ^29^Si signal level; this information was used to determine the length of wait time prior to imaging (taking into account the particles’ expected T_1_). Typical wait times ranged from 5–20 min, following which ^29^Si imaging utilized Rapid Acquisition with Refocused Echoes (RARE) sequences (α = 90°). Before moving the mouse, ^1^H anatomical imaging was acquired, also using a RARE sequence, for co-localization of the silicon signal. All imaging was performed in the coronal plane.

### 2.3. Post-Mortem Analysis of CRC Tumors

After imaging was completed, the mouse was sacrificed, and the intestine resected to remove the regions containing the tumors previously identified by digitally recorded micro-colonoscopy. Tumors were clearly evident, extending into the colonic lumen above the smooth mucosal surface at locations consistent with the imaging results. Residual particle deposition throughout the tissue and concentrated at tumors was visually evaluated and photographed. After this documentation, the excised colon was rinsed gently with buffer, and the tissue was sectioned at each of the following locations for histology: tumor, healthy colon, cecum. Hematoxylin and eosin staining was performed at the MD Anderson Division of Surgery Histology Core facility to identify MUC1-positive cancerous cells in the relevant locations. Slides were scanned using an Aperio Digital Pathology Slide Scanner with a 20X objective (Leica Biosystems, Buffalo Grove, IL, USA) and Aperio ImageScope for image viewing. Slides for immunohistochemistry were fixed in 4% (*w*/*v*) paraformaldehyde (PFA) in water. After 10 min, the PFA solution was removed, and cells were washed in phosphate buffered saline (PBS) then permeabilized for 10 min with PBS containing 0.2% (*v*/*v*) Triton-X. After blocking with 1% (*w*/*v*) BSA in PBS with 0.2% (*v*/*v*) Triton-X for 1 h at room temperature, MUC1 positive tumors were visualized with MUC1 cytoplasmic tail rabbit polyclonal antibody CT-1 at 1:25 dilution in 1% (*w*/*v*) BSA blocking buffer overnight at 4 °C.

### 2.4. Post-Acquisition Image Processing and Registration

Post-acquisition image processing is required to co-register the targeted silicon signal with the anatomical proton image. Both ^1^H and ^29^Si imaging were performed successively without disturbing the mouse’s position in the MR scanner. Imaging was performed using a dual-tuned ^1^H/^29^Si volume coil (Doty Scientific, Columbia, SC, USA), with the anatomical image taken immediately after the acquisition of the ^29^Si image. Both images were acquired using RARE sequences with a 64 mm^2^ field of view. Anatomical ^1^H images were acquired with a 256 × 256 matrix, whereas ^29^Si images were acquired with a 32 × 32 matrix. Further, ^29^Si images were acquired with a smaller matrix size due to the time-sensitive nature and spatial delocalization of the hyperpolarized signal. The silicon images are a single slice, and the ^1^H images were acquired with 22 slices (0.75 mm thickness), the most anatomically relevant being chosen for display.

Our algorithm was written to process the raw data files produced by the MRI, which included any pre-processing done through the MRI manufacturer’s software such as resizing. The data files are organized in a standard format. The code determines the correct file based on user input before proceeding. To co-register the images, the original k-space data set for silicon was zero-filled before Fourier transformation using ParaVision (version 5.1) to match the 256 × 256 matrix of the anatomical image. The raw data for all images were then imported to MATLAB (MathWorks, Natick, MA, USA) for reconstruction. This imaging procedure was performed for each of the twelve studies to compare relative signal magnitude through normalization. For each study, the silicon image was adjusted to negate signal from leftover particles outside of the mouse (typically due to the injection syringe remaining in the FOV). Each image intensity was divided by the silicon oil phantom control signal recorded just prior to injection. This normalized the silicon intensity for day-to-day fluctuations within the MR scanner.

### 2.5. Noise Reduction and Thresholding

To reduce noise, a threshold was applied to each ^29^Si image before processing together for comparison. A 32 × 32 section was taken from each of the four corners of the image, and all were averaged to establish the sample background noise statistics. After completing this for each of the twelve studies, all silicon data were compared to determine the peak signal intensity. Each image was then divided by the universal peak signal to normalize the images as a set for display and comparison. Anatomical image data were imported to MATLAB directly from ParaVision (Bruker, Billerica, MA, USA) using default settings. The relevant slice for display was chosen and saturation of the top and bottom 1% of pixel intensities was performed to enhance the contrast of the ^1^H images. The processed silicon and hydrogen images were then co-registered using a composite overlay function from the MATLAB image processing toolbox.

## 3. Results

### 3.1. Spurious Signal and Artifacts Distort Apparent Signal Intensity

We examined raw images from multiple mice to identify potential sources of spurious signal or artifacts that could alter the interpretation of findings. In Figure 1a, the signal from spilled particles (i) are located outside the mouse body. These particles are concentrated on the gauze used to support the mouse body for imaging. The density of particles creates a beacon of signal intensity that causes the internal targeted particle signal to appear weak. The syringe is left in the mouse during imaging to reduce spillage. The particles remaining in the syringe tip (panel b, ii), likewise, create a bright spot that misrepresents the signal intensity of the region of interest in the mouse GI tract. Background noise (iii) can vary between studies and apparent intensity is determined by overall signal intensity of the study. An intense signal, as seen in (panel b, ii), can cause receiver overflow artifacts (panel c, iv), due to an issue with digitalizing the analog signal.

### 3.2. Comparing Signal Intensity across Studies

In Figure 2, the difference between relative signal strength before and after processing, to account for maximum signal intensity, is shown. The left two images (Figure 2a) show that comparison of signal intensity among self-optimized studies is difficult without accounting for peak signal strength. The signal intensity is scaled based on the individual study’s absolute maximum. This can be sufficient for determining variation in signal strength for one image, but it does not intuitively portray variation among multiple images, such as might be ideal for a longitudinal or comparative study, seeking to determine if lesion progression has occurred. The two center images (Figure 2b) are the result of rescaling displayed intensity, based on the maximum signal across all of studies of interest in a comparison. The right images (Figure 2c) show the same studies after co-registration with anatomical imaging. After processing is complete, the location and intensity of signal can be evaluated among different studies, permitting more accurate image comparison.

### 3.3. Processing Artifacts Skew Displayed Intensity

Due to the low contrast of hyperpolarized silicon MR imaging, the automatically displayed image, produced from built-in processing programs, can be noisy and difficult to decipher. An example of specious signal displayed after erroneously processing images can be seen in Figure 3. If background noise is determined by a constant across studies, spurious signal will be shown, including false signal detected outside of the region where particles have been introduced. It was determined empirically, by eye, that a threshold of five times the mean background noise effectively eliminated the signal found outside of the mouse torso, as determined from the anatomical images, which could, therefore, not be attributed to the particles. After analyzing individual study background noise as a means to determine thresholding, false noise was reduced successfully, as shown in Figure 3c,f.

### 3.4. Quantitative Analysis Shows Processed Signal Registers to Tumor Location

To quantify the differences seen in targeted vs. control studies and to show the specificity of signal relative to tumor location, analysis using ImageJ was performed to determine signal-to-noise (a) and contrast-to-noise ratios (b) (Figure 4). The mouse torso ROI was used as control (c). Each tumor location (d) was indicated on an anatomical image slice using information from the anatomical images, in conjunction with relative tumor location, determined by prior colonoscopy. The exact area of tumor locations was shifted to non-tumor regions (e) for comparative analysis. Signal-to-noise ratio was calculated using the below equation
SNR = |μ_t_ − μ_b_|/σ_b_,
where µ_t_ is the mean intensity at the position of the tumors, µ_b_ is the mean intensity in a region away from the tumors, and σ_b_ is the standard deviation in a region away from the tumors, with all values calculated using a standard 7 mm diameter ROI, based on visual determination. Contrast-to-noise ratio was calculated similarly as
CNR = |μ_t_ − μ_m_|/σ_m_,
where µ_m_ and σ_m_ are the mean and standard deviation, respectively, of the measured signal in the mouse torso, excluding the tumor regions. The average SNR between targeted and control groups had *p* < 0.001 and the average CNR between targeted and control groups had *p* < 0.05, indicating strong supportive evidence for positive targeting ability. Statistical analysis was performed with analysis of variance, using GraphPad Prism (San Diego, CA, USA).

### 3.5. Post-Processing Reveals True Signal

Figure 5 shows the significance of post-processing on the displayed intensity of diagnostic imaging files. The raw image file before processing is noisy and signal intensity is skewed by particles left in the syringe, as well as particles that have leaked post-administration (Figure 5a). The post-processed image shows a meaningful signal after denoising and disregarding the false signal from outside of the mouse (Figure 5b). The location of tumors was correlated to areas of high signal post-processing. The background signal was determined by averaging the signal intensity of the four corners, away from the physical mouse location. The background for an individual study is determined before any further processing occurs and the intensity value is maintained after the image is visually altered for display purposes. However, there could have been signal outside of the mouse that should be ignored due to setup or experimental physical limitations. To avoid including erroneous signals, a quadrangle was drawn to indicate the ROI, though the image could also be processed without specifying an ROI, without complication. The peak intensity for a study in the user-defined ROI was determined and recorded for comparative representation. The result after processing was an image largely clear of artifact, with signal aligned with visually identified tumors.

## 4. Discussion

### 4.1. Distortion of Signal Due to Artifacts Compromises Meaningful Signal

Various sources can result in artifacts that distort the automatically displayed signal image. Imaging methodology can produce spurious signals that misrepresent the true targeted signal. Whether the result of MRI artifacts or experimental set up, false signal can misrepresent signal intensity, leading to incorrect or misinformed diagnostic assessment. One of the most common artifacts in hyperpolarized ^29^Si imaging is RF overflow. Despite the long longitudinal relaxation time constant (T_1_) of silicon particles, the hyperpolarized state is transient and relaxes to thermal Boltzmann equilibrium. As hyperpolarization can lead to over 10,000-fold signal enhancement, the initial signal has a tendency to create RF overflow artifacts that led to a non-uniform, washed-out or echo appearance in an image, as seen in Figure 1c. This artifact occurs when the signal from the hyperpolarized silicon particles, received from the amplifier, exceeds the dynamic range of the analog-to-digital converter, causing clipping [23,24]. In conventional MRI, autoprescanning usually adjusts the receiver gain to prevent this overflow artifact from occurring, which is not possible in the case of hyperpolarized MRI. Through experience, we have learned to decrease the receiver gain manually in hyperpolarized MRI but overflow still happens often. As with conventional MRI, this can be corrected by post-processing methods [25].

### 4.2. Cross-Study Importance of Evaluating Relative Signal Strength

A more quantitative method for assessing these images is by post-processing, using cross-study signal intensity as a baseline for image comparison. Before relative intensity is considered, the signal observed in these individual studies appears to have equal maximum intensity, where they cannot be differentiated in a comparative study. After considering and correcting for the maximum intensity across studies of interest, the relative signal strength in each study becomes clear, as seen in Figure 2. The studies with previously high signal appearance are reassessed to display a more accurate representation. This correction is important, both for longitudinal studies, where a region of interest is reimaged over time, and for dynamic studies, where changes in signal intensity within a region are determined by imaging multiple times within a single imaging session. In addition, by setting the threshold of background noise to a generic variable, user error is minimized.

Most high-risk individuals in need of noninvasive MRI-based colonoscopy will not have one procedure in a lifetime, rather, they would need to be evaluated at periodic intervals. When image processing is optimized on a case-by-case basis, at a single time point, the ability to compare lesions is lost. The processed image will be self-normalized, potentially resulting in variation in the signal intensity representation if backgrounds differ. Because each image is analyzed and processed individually, comparative conclusions cannot be drawn between images. This limitation is an issue for pre-clinical studies, where comparison is useful to determine the significance of specificity against controls. This shortcoming is also detrimental for situations where lesion size needs to be tracked over time. An example of this would be to track treatment response, where imaging is done periodically to determine the efficacy of a current treatment plan in reducing tumor size.

### 4.3. Artifact’s Impact on Displayed Signal

MRI artifacts are inherent in imaging methods with lower signal to noise. These artifacts can be intensified based on the post-processing method. In preclinical studies using mice, if the region of interest is not specified, a spurious signal localized outside of the mouse torso may appear as significant. Additionally, MRI artifacts can further complicate the visually displayed significance of the apparent signal. This issue can be addressed by accounting for individual study noise background. By taking the average of the four corners away from the region of interest, each study can be assessed based on the specific noise during the time of acquisition.

### 4.4. Post-Processed Images Reveal Tumor Location

False positives in cancer diagnostic imaging can result in unnecessary treatments and negative outcomes, in addition to avoidable financial and emotional burden for the patient [26,27]. Additionally, a false positive result increases cost of care due to unnecessary follow-up screening, which can be invasive [28]. False negatives can result in a delay of appropriate treatment, leading to poorer prognosis. False positives and false negatives can be the result of the diagnostic method, imaging sensitivity, or human error. The algorithm described here decreases false diagnosis by using automated, unbiased post-processing. The results of this methodology on the example studies showed zero instances of false negatives and fewer instances of false positives, assessed as residual bright spots, when compared to previously used methods in our laboratories. The difference in signal between tumor location and background was significant between the targeted studies and the controls. Statistical analysis supports the accuracy of this post-processing algorithm and the capacity to reveal true signal, expected to reduce the number of false diagnoses.

### 4.5. Algorithmic Processing Indicagtes Relative Signal Intensity across Dynamic Studies

The original processing code was written using MATLAB, which contains proprietary tools and is less accessible due to cost and availability. MATLAB generally requires a deeper level of coding background, hindering its usefulness as a tool in medical practice. Python is free and open source, with a large online community. Source code can be hosted and shared freely. Additionally, Python is becoming one of the most desired and widely known coding languages [29]. This makes it an excellent choice for applications where users may not have a strong background in coding. It can be utilized by anyone, regardless of institution-provided subscription or access to funds.

The initial Python co-registration algorithm implementation required multiple script-level modifications for proper execution. In order to make the image viewing simpler, a Python Graphical User Interface (GUI) was written to encapsulate this dense back-end software. Using PyQt, a port of the popular cross-platform Qt window development suite, the resulting GUI requires only a simple Microsoft Excel configuration file. All other user-interaction takes place through intuitive input formats, such as buttons and text-input fields. This open access code can be used to process any imaging data where co-registration and comparative analysis is desired. Because the code is open source, users can also make modifications to fit their own specific research needs. This method of processing allows the viewer to make qualitative comparative analysis of time dynamic studies, as well as longitudinal studies. This method is almost fully automated and allows the user to restrict the region being analyzed if desired. Full documentation and instructions can be found at https://github.com/farachcarsonwulab (accessed on 22 April 2021).

## 5. Conclusions

Noise and other artifacts complicate the readability of MR images for clinicians on site. The automatically displayed image cannot be edited beyond superficial individual adjustments, such as windowing and leveling. This results in images with varying intensity that may be affected by signal sources outside of the main region of investigation. However, the processing algorithm presented here permits users to determine the appropriate background removal, on a case-by-case basis, without user interference for individual assessment. Background noise can be determined based on a universal SNR threshold, set by the user to accommodate all images of interest, to elucidate the relevant signal of interest. This algorithm aims to decrease the number of false diagnoses, while increasing the accessibility of code modification for application-specific post-processing customization.

## Figures and Tables

**Figure 1 diagnostics-12-00610-f001:**
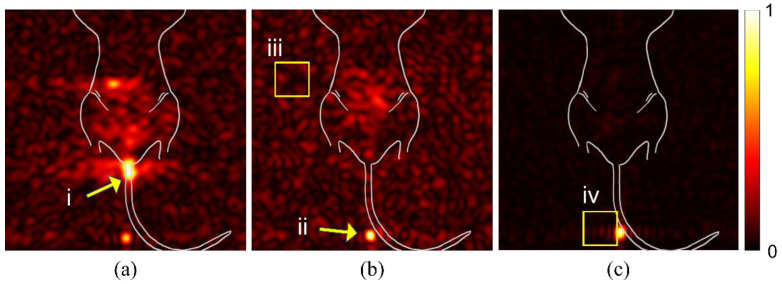
Spurious signal and artifacts distort apparent signal intensity. Three in vivo studies (**a**–**c**), with overlaid illustration of mouse anatomy, show examples of issues and artifacts that misrepresent relevant signal for targeted studies. (i) Spilled or leaked particles show concentrated signal outside of the mouse. (ii) Leftover particles in syringe distort the signal intensity of targeted particles. (iii) Background noise due to day-to-day variations with the MRI can distort expected signal-to-noise ratios. (iv) RF Overflow artifacts can cause distortion in visualization of signal strength.

**Figure 2 diagnostics-12-00610-f002:**
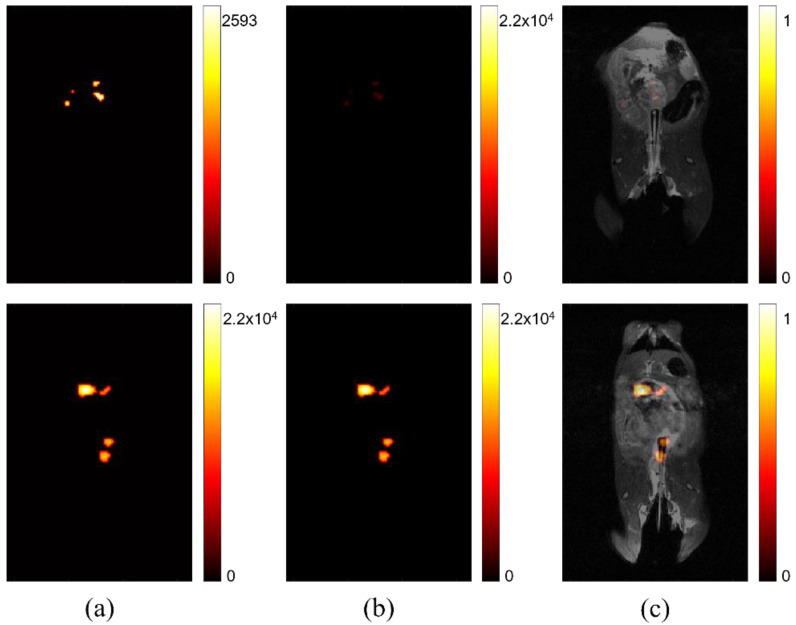
Comparing signal intensity across studies. (**a**) Two separate studies where maximum signal intensity is displayed according to the highest value for each respective study. (**b**) The same studies after post-processing to display maximum signal intensity normalized using the highest signal measured across all studies after individual normalization of static background per study. (**c**) The adjusted signal intensity after co-registration with anatomical hydrogen MRI.

**Figure 3 diagnostics-12-00610-f003:**
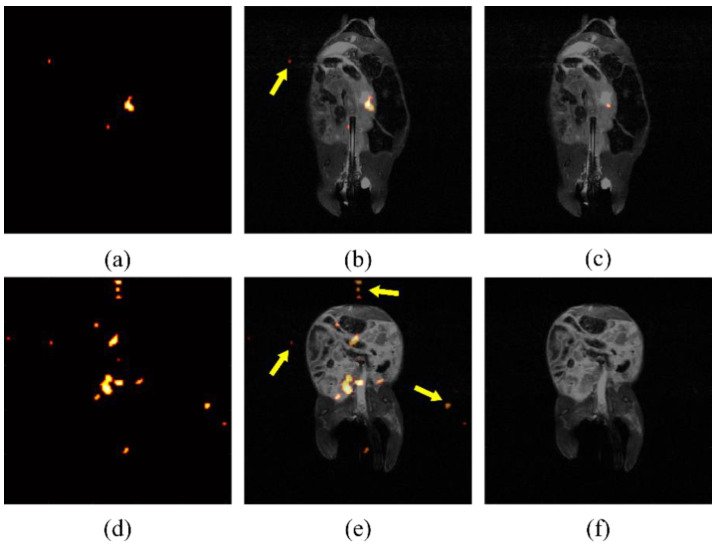
Processing artifacts skew displayed intensity. False signal in raw silicon images from two mice (**a**,**d**), indicated by arrows superimposed on anatomical image (**b**,**e**), is still present after using a standard threshold for noise reduction. Noise intensity may change between studies due to day-to-day variation in the magnetic field. After normalization for noise levels present for individual studies, the false signal is largely eliminated (**c**,**f**).

**Figure 4 diagnostics-12-00610-f004:**
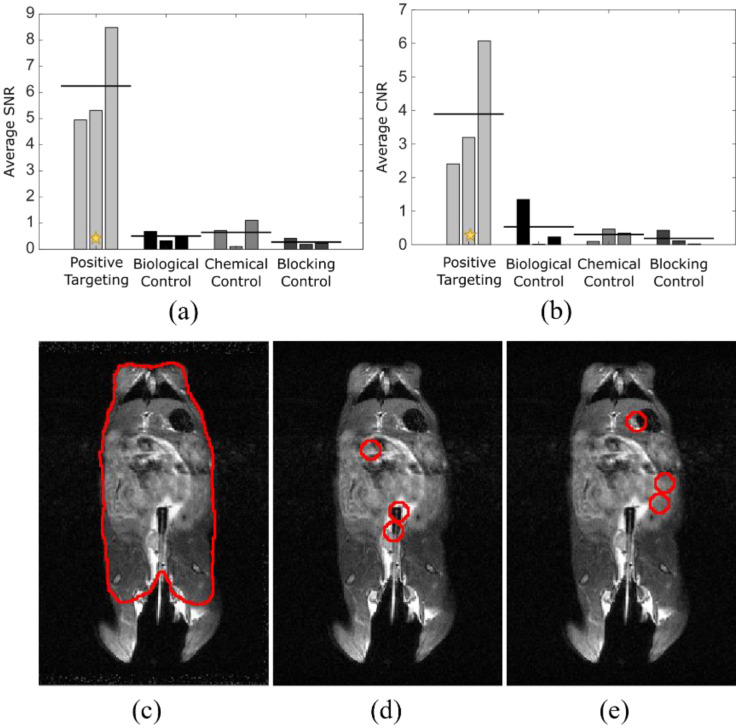
Quantitative analysis shows processed signal registers to tumor location. Analysis of SNR (**a**) and CNR (**b**) for four experimental groups using ROIs drawn for mouse body (**c**), tumor locations (**d**), and non-tumor locations (**e**) of the same area as those from (**d**). Tumor locations were determined with anatomical MRI and were confirmed following excision of the colonic tissue. *n* = 3 for each cohort. The star indicates the example study (**c**–**e**) data point after analysis [22].

**Figure 5 diagnostics-12-00610-f005:**
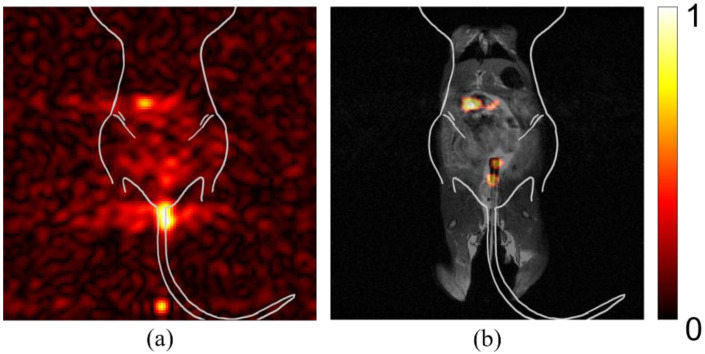
Post-processing reveals true signal. Mice were anesthetized prior to being imaged in a 7 T small animal research MRI. Imaging was performed using coronal slices with the mouse in a supine position. (**a**) Pre-processed silicon signal. Spurious signal obfuscates ability to detect true tumor sites. (**b**) Post-processed image of the same study, co-registered with anatomical imaging. True signal becomes evident after processing to reduce artifact-associated signal. Color bar in arbitrary units of highest intensity.

## Data Availability

Not applicable.

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
