# Peer review of "Post-Acquisition Hyperpolarized 29Silicon Magnetic Resonance Image Processing for Visualization of Colorectal Lesions Using a User-Friendly Graphical Interface"

_diagnostics, 2022, doi:10.3390/diagnostics12030610_

Round 1
Reviewer 1 Report
Congrats for this very interesting article. It's a good article with interesting findings. An article whose findings are important to those with closely related research interests.
The paper is well written and correct from a methodological point of view.
I have only a minor recommendation: preferably do not use abbreviaitons in title.
Author Response
Point 1: I have only a minor recommendation: preferably do not use abbreviaitons in title.
Response 1: We thank you for the enthusiasm shared for our work. We have removed the abbreviation “MR” from the title and replaced it with “Magnetic Resonance” as suggested.
Reviewer 2 Report
Thank you to authors to sending the articule “Post acquisition hyperpolarized Silicon
MR image processing for visualization of colorrectal elsions using a user-friendly
graphical interface”.
The article is interesting and opens the door to new diagnostic methods of colorectal
lesions. Using antibody-targeted molecular imaging in a mouse model, authors
described a new methodology that allows users to calibrate a set of images acquired
with MRI and locate colorectal tumors in living mice. However, certain improvements
can be made to facilitate your understanding.
Please write the article according to authors guidelines and order the sections on
introduction matherials and methods, results, discussion and conclusión.
References might be rewrited according to authors guidelines.
Institutional Rewiew Board Stament should include the protocol number.
Introduction is to large, it might be shortened.
Descricption of the post-mortem analysis of CRC tumors should be more extensive to
facilitate compression of the correct method of identification of cancer cells.

Author Response
Point 1: Please write the article according to authors guidelines and order the sections on
introduction matherials and methods, results, discussion and conclusión.
Response 1: We very much appreciate the enthusiasm expressed for our work. In response to the excellent suggestions, we have proofed the article according to authors guidelines and ordered the sections as suggested to include the introduction, materials and methods, results, discussion and conclusions.
Point 2: References might be rewrited according to authors guidelines.
Response 2: We have proofed and edited the references to align with journal guidelines.
Point 3: Institutional Rewiew Board Stament should include the protocol number.
Response 3: We have added the IRB approval code and approval date to the manuscript.
Point 4: Introduction is to large, it might be shortened.
Response 4: We thank the reviewer for pointing out the length of the introduction and agree that shortening it will improve this manuscript. It has been ammended to be more concise.
Point 5: Descricption of the post-mortem analysis of CRC tumors should be more extensive to
facilitate compression of the correct method of identification of cancer cells.
Response 5: We agree and appreciate this being pointed out. We have rewritten this section with greater detail regarding the post-mortem analysis of the CRC tumors to clarify how cancer cells are identified and distinguished from normal cells. In particular, the staining and imaging procedure of the tissue sections are now included to improve comprehension and reproducibility.